# Female Labor Participation and Mother's Daycare in Greece Pre- and Post-COVID-19 Pandemic: A Regional Analysis

Costas Siriopoulos [1],*  and Sophia A. Kassapi [2]

1   College of Business, Zayed University, Abu Dhabi P.O. Box 144534, United Arab Emirates
2   Department of Business Administration, School of Economics and Business Administration, University of Patras, 26504 Patras, Greece; skassapi@upatras.gr
*   Correspondence: konstantinos.syriopoulos@zu.ac.ae

**Abstract:** This paper studies the sample of 1,177,794 mothers in Greece who were eligible to apply for and receive a voucher as beneficiaries of the European program *Harmonization of family and business life* for the period 2011–2022, using financial on/off criteria. This was a European union (EU) subsidized program under the main strategic plan for Growth in Greece, the Partnership agreement for the Development Framework. This paper conducts an ANCOVA analysis to reveal the motives of the participants that led to the progress of the program both pre- and post-COVID-19 pandemic. The database comes from all applications submitted online, but confidential data remained undisclosed, restricting the power of the results. Findings uncover the major factor interfering with a mother's decision to apply prior to COVID-19, namely her family status. However, for the post-COVID period, a mother's declining age is the major determinant, which shows the increasing uncertainty of women in reproductive age. These findings, along with the low birth rate in Greece, suggest large social unrest, such that a large fraction of women in the workforce would go into labor if they had greater access to subsidized childcare. Regional differences do not seem to affect these results. This analysis might be unique in its nature, as no previous paper has focused on mothers' characteristics linking them to childcare.

**Keywords:** Greece; subsidized daycare; gender equality; welfare programs; fertility; ANCOVA

**JEL Classification:** R23; P25; O43; G23; I38; J13

## 1. Introduction and Motivation

The work of this paper builds upon women who live in a continuous relationship (marriage or cohabitation), having added the status of families with one parent, since the beginning of the program [1,2], from the 2010–2011 (pre-COVID-19 period) to the 2021–2022 (after-COVID-19 period) school year. It also pays special tribute to those suffering the repercussions of the effects of the COVID-19 pandemic on families, which had a direct effect on the progress of the program, as revealed by the data at hand.

The spread of the COVID-19 virus has changed how we lead our lives, how markets and governments/incumbents react, and how political leaders are making decisions at the local, national, regional, and global level. Most research on the political effects of natural disasters [3], political instability [4], terrorism, capital controls, and economic/financial shocks [5] have focused on their impact on economic growth and finance development, on contagion effects as the cost of the globalization, etc. The same questions are investigated in recent COVID-19 pandemic research papers. Some authors have investigated the key factors in cross-country heterogeneity regarding economic activity during the last pandemic [6] and assessed the support given to the markets by relevant governmental actions. Other studies have started to assess the effects and risks of the COVID-19 pandemic [7,8], and others have questioned its effect at the regime level [9,10], and with regard to democracy [11,12],

while others report a downward trajectory in political trust [13]. However, no studies have discussed how the role of female labor participation and mothers' daycare has been affected in the post-COVID-19 era. In the present paper we aim at providing evidence to this question across the different Greek regions, in reference to the European program financed by the European Social Fund, National Funds and Ministry of Labour and Social Affairs Funds. This analysis might be unique in its nature, as no previous paper has focused on mothers' characteristics that links them to childcare.

The provision of daycare to women for their offspring is a measure strongly linked to growth and development for society as a whole. Gender equality at work has been on the agenda for the past few decades as a key driver of the protection of human and civil rights. The United Nations (UN, [14]) makes imperative the ending of poverty in all its forms and European Union (EU) policy toward improving childcare, clearly set up since the 2002 Barcelona meeting to remove "barriers and disincentives for female labor participation by, inter alia, improving the provision of childcare facilities", asks for women to be allowed to work and make their human capital available in the labor market.

Female labor supply and fertility is an important issue for our modern economies for several reasons. Following the trend in the literature on female labor supply and fertility [15], this study is mainly based on the panel data from the Hellenic Establishment for Regional Development & Administration [16]. Data were taken from https://www.eetaa.gr/ (accessed on 13 March 2020 and 2 September 2022). EETAA [16] is the official institution responsible for the implementation of this program, co-financed by the European Social Fund, National Funds and Ministry of Labour and Social Affairs Funds. The program *Harmonization of Family and Work life* was implemented under this name until 2021–2022 for two consecutive programming periods of the (NSRF) National Strategic Reference Framework, 2007–2013 and 2014–2020. This action is being continued in the 2021–2027 NSRF, handled by EETAA under the name *Program for preschool learning and children's leisure activities.*

The paper focuses on women who live in a continuous relationship (marriage or cohabitation), adding the status of families with one parent, from the beginning of the program, from the 2010–2011 (pre-COVID-19 period) to 2021–2022 (post-COVID-19 period) school year. This paper focuses on the effect of the COVID-19 pandemic on families, which has also had a direct effect on the progress of the program, as revealed by the data at hand.

The motivation for our research is based on various facts regarding participation in the program pre-and-post-COVID-19 pandemic, the average age of women giving birth, and on the asymmetric involvement across different regions of Greece. These facts are made clearer to the reader by the following Figures 1–4. It is important to note that interest in the program showed signs of decline (Figure 1) in the school year 2019–2020, just before the COVID-19 pandemic. The increasing trend in applications is interrupted in the school year starting September 2019 before the COVID-19 pandemic; however, it increased sharply during and after the COVID-19 outbreak. This can be explained by the increasing uncertainty post-COVID-19.

A second observation, very revealing for this study, is the differentiation in applications across the 13 Greek regions, as is shown in Figure 2. In regional terms, the interest in the program throughout the years was, as expected, large in the urban and rural areas of the region of Attica (Region 9), the most populated region and home to the country's capital. An even bigger interest arose in the region of central Macedonia (Region 2), and this was noticeably greater. Central Macedonia is the second biggest region, accounting for 10% of the population.

A key fact is that mothers interested in applying for subsidized daycare from the program showcase an advanced maternal age compared to the average age of a woman giving birth for the first time, taking into account the age of the child. For example, for the school year 2021–2022, the mean age of a mother participating in the program and bringing her child to daycare is 38. (The age of a mother is calculated as shown at the time of the application. Data does not reveal the exact age of the child as it is not a prerequisite when

applying. We estimate the age of the child between 0–3 years.) The country's corresponding average mother's age is 30.6 years.

As shown in Figure 3, the average age of a woman giving birth to her first child is ≥30 for Greece. In times of crisis, the determinants of fertility dynamics cause people to postpone or revise their fertility plans, potentially serving as an indicator of better upbringing of fewer children (Eurostat, 2021). In Figure 4, the mean age at childbirth is steadily rising, and is quite higher than the EU average. Therefore, the possibility of having more than one child is diminishing.

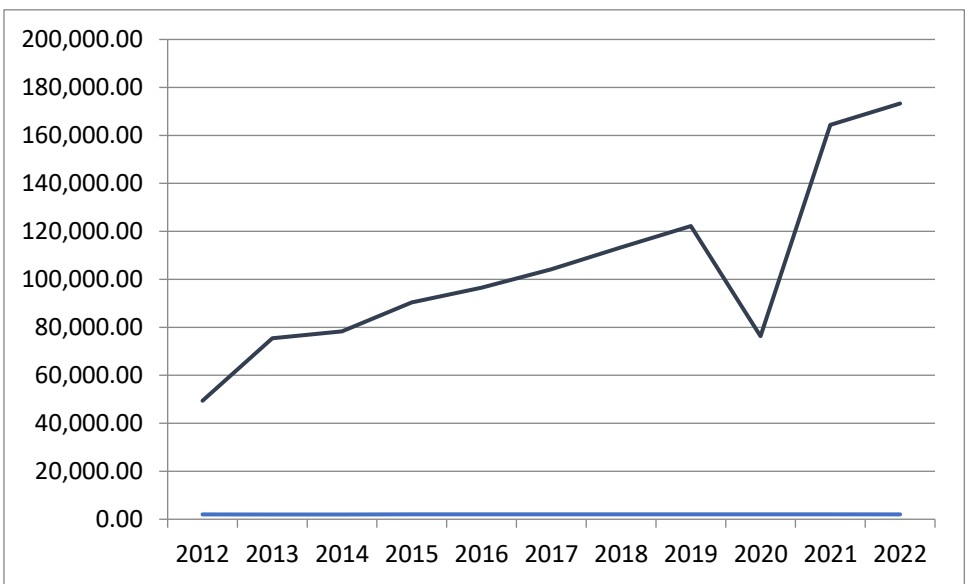

**Figure 1.** Figure 1. Pre-pandemic and post-pandemic phases of the program. Number of Applications per year. Source: Data taken from EETAA database.

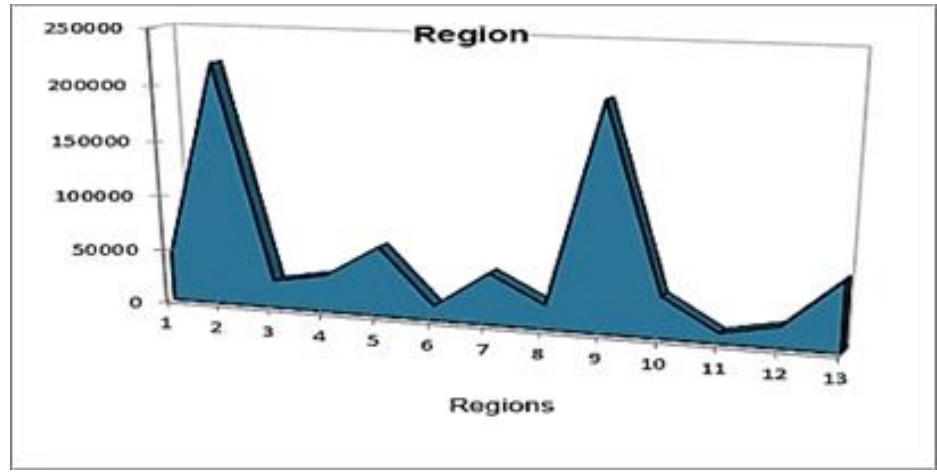

**Figure 2.** Applications per region total. Source: Data taken from EETAA database.

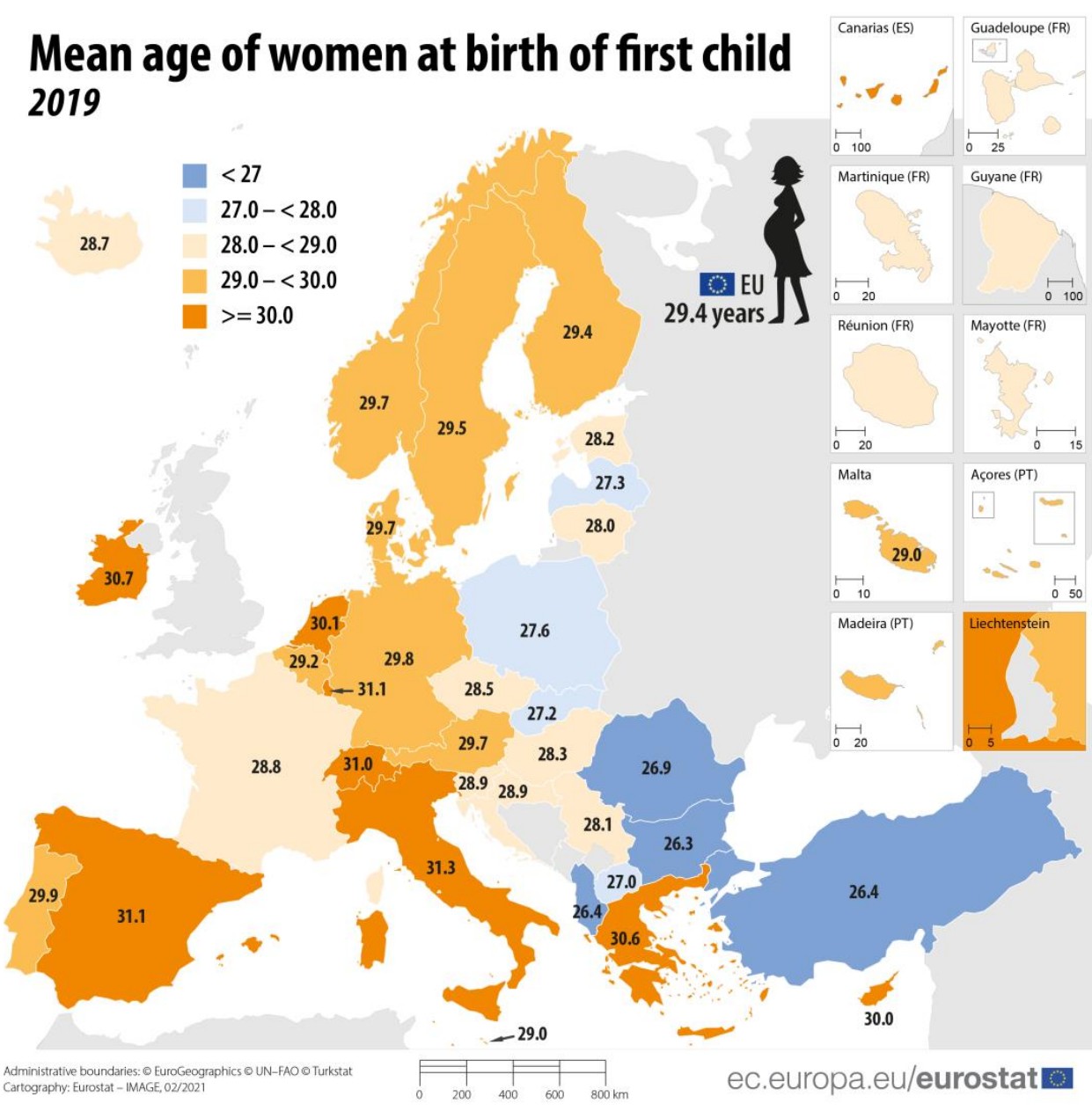

**Figure 3.** Mean age of women giving birth in EU Reprinted/adapted with permission from Ref. [17] 2023, Eurostat (2021) (https://ec.europa.eu/eurostat/web/products-eurostat-news/-/ddn-20210224-1 (accessed on 1 February 2023)).

Without urgent action, the number of children living in poor households across low- and middle-income countries shows a stochastic increase by 15%, aka 672 million children in 2020. Save the Children, together with UNICEF [18], revealed this fact, focusing on the effect of the pandemic on children. For Europe, the estimates predicted a surplus of up to 44% across the continent. There is an immense need to provide greater support and gender equality to new parents. Guaranteeing more access to work and economic development for women and reducing barriers for fathers to take time off from work in case of birth or adoption could be a sustainable measure to overcome this crisis [17]. The COVID-19 pandemic created a childcare crisis and mothers bore the burden. Childcare is about economic justice for women. Having to face the COVID-19 pandemic, some governments invested in other forms of social protection, fiscal policies, employment, and labor market interventions, and in family friendly policies, such as paid leave and childcare, to

support families (For instance, in Australia, see https://budget.gov.au/2022-23-october/content/cost-of-living.htm#fvm2 (accessed on 1 February 2023). In Canada, https://pm.gc.ca/en/news/news-releases/2021/04/21/10-day-child-care-canadian-families, and in Ireland, https://www.employersforchildcare.org/news-item/irish-government-launches-major-reform-of-funding-for-early-learning-and-childcare-sector/ (accessed on 1 February 2023)).

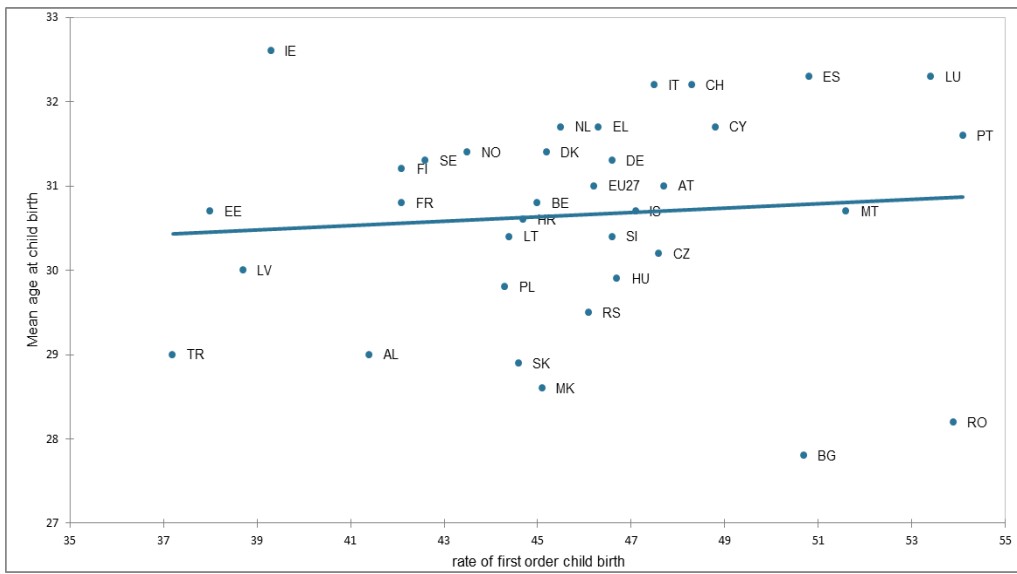

**Figure 4.** Fertility indicators. First order child rate and mean age of women at childbirth in Europe for 2020. Data taken from Eurostat (Last update 9 November 2022).

Beneficiaries of this subsidized welfare program are given the chance to enter the labor force and acquire some training during their children's stay at daycare. A mother that can create stimulation and acquire resources for a child is normally a woman who is educated and can work and earn more. What this program aims to accomplish is to raise the quality of parenting, as it is seen as a measure of a more or less disadvantaged environment for children [19]. For a mother to provide a healthy environment, it would require more quality time devoted to child development and the provision of a stable family at the same time. The authors [20] suggest that, for a family to grow with respect to less accomplished women, financial and cognitive resources and intact social behaviors should be present. It was shown that gaps in ability emerge early and persist. These ability gaps that make their presence felt at age 18 are also present at age 5, and this more or less explains all the gaps in adult outcomes. Schooling further on does not play the leading role in creating or extending these gaps, as [21] shows. American children with very different family backgrounds bearing an odd resemblance in their test scores at parallel age implies the presence or lack of proper government planning towards preschool education and care.

More mothers would increase their earnings and seek new job opportunities if they had greater access to reliable and affordable childcare, as many families must make a choice between spending a significant portion of their income on childcare or leaving the workforce. In a recent study, it was estimated that two million parents in the USA made career sacrifices due to problems with childcare [22]. It is important to study the effect of a European childcare program, its implementation, interest in it, and its performance in a small European economy with financial problems, such as Greece.

Our paper differs from previous examples, as it focuses on mother's characteristics that drive them to childcare.

Most previous studies on the quality of childcare concerning parents' and teachers' perceptions conclude that reform is imperative for the unification of the two parallel systems of childcare.

For the education system of Greece, no-one can define whether the end of childcare is the beginning of education and development, and this is reported to be the reason behind this "split" model of two parallel and distinct structures [23] available on the ground. A dichotomy makes all the difference in the way the provision of preschool education has been designed. Preschool education and childcare are two different things provided separately, even though we are talking about two concepts which are bound together.

Ref. [24] conducted a correlation analysis on a sample of 141 parents who had their children enrolled in 27 classes in 18 childcare centres in the Ioannina region of Greece, evaluating the mechanisms deployed in the quality and care provided by the Greek preschool programs from the researcher's and parent's perspective and found that parents evaluated comparatively higher those who had their children enrolled in public programs.

Ref. [25] studied the quality of the student-teacher relationship using the Student-Teacher-Relationship-Scale (STRS), a 28-item questionnaire assessing three dimensions, namely closeness, conflict and dependency, and using an Exploratory structural equation modeling (ESEM) technique across 12 areas of Greece. They used a random sample of 535 early education classrooms and 535 educators from both parallel structures, including eight children, and found a positive correlation between closeness and dependency, confirming the existence of cultural differences in teacher–child relationships, with females being more close and less conflictual in comparison to males.

Ref. [26] studied the quality of early childhood education and care services (ECEC) in Northern Greece, using a sample of 131 daycare classrooms, 98 in urban and 33 in rural areas, with the Infant-Toddler Environmental Rating Scale Revised Edition (ITERS_R). The findings suggested the existence of average quality and very low scores on parents' involvement, acceptance of diversity, and supervision and evaluation of staff. They suggested that the current economic crisis augmented by the constant influx of immigrants and the demographical changes of the recent past dictate the need for clear cut measures for more resilient childcare.

This contemporary study analyzes and clarifies the priorities parents themselves set out the moment they choose to send their children to an early childcare facility.

The paper continues in the following order: Section 2 provides the characteristics of the data sample that is being used for the purpose of this analysis. Section 3 is a brief discussion of the results, examining and evaluating how much this study offers to the early childhood education framework. Finally, Section 4 concludes the paper.

## 2. Materials and Methods

In 2010, the Greek government, along with the European Union, initiated a major co-financed childcare program across all regions and kindergartens in its territory. Regarding this expansion in subsidized childcare, throughout its analysis, this paper will shed light on important characteristics of financially struggling groups in society who, against all odds continue with planning the lives of their family. Data is drawn from the second wave from 2012 to 2022, spanning the years 2011 to 2021, because the variables in child registration refer to the beginning of the academic year, and income to the year prior.

ANCOVA (Analysis of Covariance) is a model that has both qualitative and quantitative independent variables and has been performed in Excel with XLStat software. The results contain residuals analysis, Type I SS, Type III SS, and predictions. Using ANCOVA, this paper tests which of the independent variables of our sample has the most effect on the dependent variable, mothers; income, which is also the major determinant of the applicant. Therefore, this will reveal which of the explanatory variables and factors displays a positive relationship and leads the sample. By using a GLM model, the relationship we are looking for is currently linear.

### 2.1. Sample Selection and Data Cleaning

The analysis of this paper is based on the panel data from the European early childhood program facilitated by The Hellenic Agency for Local Development and Administrative

Regions, *Greek local development and Administrative Regions* (EETAA, https://www.eetaa.gr/ (accessed on 22 January 2023)). Figure 5 depicts the map of Greece along with the 13 Administrative Regions. Greece has an interesting landscape with a large number of islands, and most income comes from the tourist industry. In addition, the economic differences between rural and urban regions are large. Hence, the response rate to the particular EU program is expected to be differentiated across the various regions.

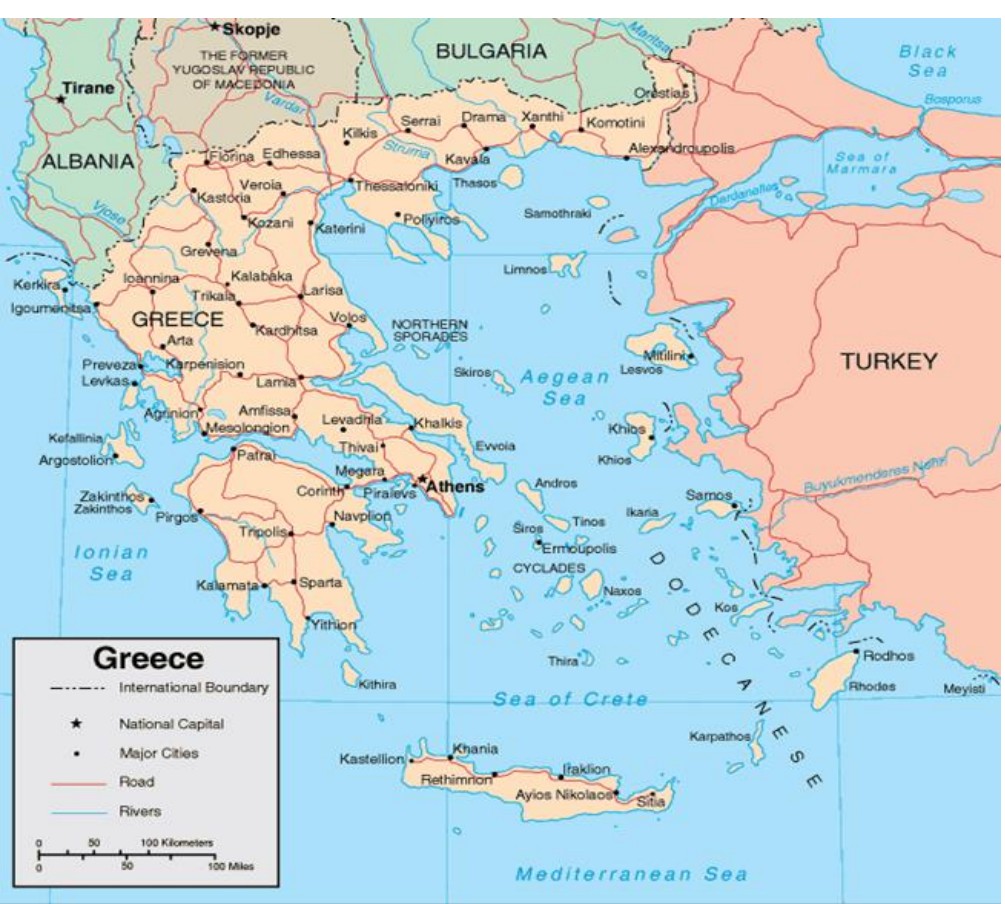

**Figure 5.** Administrative regions of Greece (see Appendix A, Table A1).

This is the only official database, containing reliable statistical data, in Greece [27]. Its setting up was aimed at the monitoring and coordination of daycare centers that participated in the "voucher" program, comparable in scope to other European programs aiming to provide subsidized childcare in other EU member states. The variables in labor force participation and income that are used go back to the year prior to the facilitation of the program, as do the criteria set for those interested in applying.

Following the common practice in the literature on female fertility [28–30], the age of the mother is considered to be between 15–49, allowing values up to the age of 55. This is due to two reasons: women give birth at ages above 50 currently, and this will be encountered. In addition, the starting age of the child who has the right to receive early childhood education and care was 6 months old (until 2017) or 2 months old (after 2017). The age of a mother is calculated as shown at the time of the application. Data does not reveal the exact age of the child, as it is not a prerequisite when applying. We estimate the age of the child as between 0–3 years. However, most working mothers are entitled to maternity leave and bring the child at the minimum age of one, or at least during that year. The right to attend childcare extends to the year when the child has its fourth birthday, therefore in its fifth year, and we allow another 5 year period for the mother's age from birth, adding one more year for the legal limit of right to register to childcare, up to the age of 55, aiming to contain all age groups in the sample.

The program has received a total of 827,558 applications during the pre-COVID 2019–2020 period and 348,236 for the post-COVID 2020–2022 period we are controlling for. After January 2020 (when WHO declared a public health emergency of international concern) and until January 2021, a total of 240,000 applications were recorded. A number of 21,296 applications were removed from the 2011–2020 sample and 10,484 from the 2020–2022 sample, respectively. The reason is that these applicants were given access to the program for completely different reasons that are not being examined here. That is, they were either parents of children with special needs having the right to attend daycare or evening center for children's leisure activities (KDAP) without any income restrictions, or legal guardians within a very different age range, such as a grandparent or any other caretaker/relative of those children who for any reason were not living with their parents. These groups of applicants deserve to be included at a later stage in another analysis with a broader range of childcare provision.

The analysis focuses entirely on women living permanently on Greek soil, either as citizens or with a legal citizen permit, therefore we have merged citizens from different European countries as "EU citizens" and those applying with a legal work permit from other countries outside the EU as "non-EU citizens". Women employed as public officers with any type of employment (temporal, permanent, etc.) are excluded from this program at this phase. Labor force participation status of mothers has been set according to whether a mother is working full-time, part-time or occasionally. Employment status reflects how much a mother has worked in total during a period in her child's adolescence.

As the applications for this program were submitted via a local government agency, applications are categorized and segmented in each one of the 13 regions of the country, so that one can get a closer look at what is going on at the regional level, as well as in terms of a mother's age, occupation, income, etc.. All aggregate annual statistics on the provision of childcare are made available from Eurostat.

### 2.2. Variable Definitions and Method

The main dependent variable is mother's income (MINC). Father's income (FINC) has been chosen as one of the covariates. Mother's age (MAGE) is an ordinal covariate, and mother's nationality MNAT, mother's work status MWORK, mother's family status MFAM, mother's education level MED and the respective region of residence REG are the factors included in the model. Having chosen to control for all factors affecting the participation in the program and the family income, MINC and FINC are both examined as dependent and covariates of one another in univariate models. Other variables in the sample regarding nationality, work status, family status and mother's education level displayed some common characteristics across regions. Most mothers were of Greek origin and married. In addition, with an equal and rather large share of the whole population, mothers were either unemployed or employed full-time. A small proportion ran a business on their own or was occupied part-time (those in urban areas were hired through programs, and those in rural were mostly occupied with land activities).

ANCOVA analysis assesses whether the means of the dependent variable (MINC) are equal across levels of the "covariates" (FINC and MAGE), while statistically controlling for the effects of other categorical variables (MNAT, MWORK, MFAM, MED and REG). Hence, we decompose the variance of MINC into variance explained by FINC and MAGE, variance explained by MNAT, MWORK, MFAM, MED and REG, and residual variance.

All variables in the sample, e.g., nationality, work status, family status and mother's education level, displaying common characteristics across regions are depicted in Table 1. Most mothers are of Greek origin and married. In addition, with an equal and rather large share of the whole population, mothers are either unemployed or employed full-time, and there are no working women in part-time or on-the job training programs. Only a small proportion run their own business or were occupied part-time (those in urban areas were hired through programs, and those in rural areas were mostly occupied with land activities). Information regarding their education level shows that most are high school graduates,

while very few have obtained a graduate degree. Regarding the pre- and post-COVID-19 samples and the age of the mother, a mean of 36 years old for the first sample was replaced by one of 38 years old for the second, post-COVID-19 sample. Hence, as uncertainty increased in the after COVID-19 period, the program attracted more applications from aging mothers (Figure 6).

**Table 1.** Variable type, definition and period under study.

| Variable Name | Variable Type | Definition/Source | Period |
|---|---|---|---|
| MINC/Quantitative | 0–40,000 Euros | Mother's income/EETAA | 2010–2022 |
| FINC/Quantitative | 0–40,000 Euros | Father's income/EETAA | 2010–2022 |
| MAGE/Quantitative | 15–55 yrs. | Mother's age/EETAA | 2010–2022 |
| MNAT/Qualitative | Greek, EU, non-EU | Mother's nationality/EETAA | 2010–2022 |
| MWORK/Qualitative | Employed, unemployed, self-employed, in-training * | Mother's work status/EETAA | 2010–2022 |
| MFAM/Qualitative | Married, Non-married, Divorced, Widow | Mother's family status/EETAA | 2010–2022 |
| MED/Qualitative | Tertiary, Non-tertiary, N/A | Mother's education/EETAA | 2010–2022 |
| REG/Qualitative | 13 regions | Region/EUROSTAT | 2010–2022 |

\* The categorization of mother's work status changed in the post Covid-19 sample and was more analytical concerning the part time jobs and training programs, thus for the data to be comparable in scope we kept the categorization of the data since the beginning of the program which served quite well as the four major occupation categories for mothers.

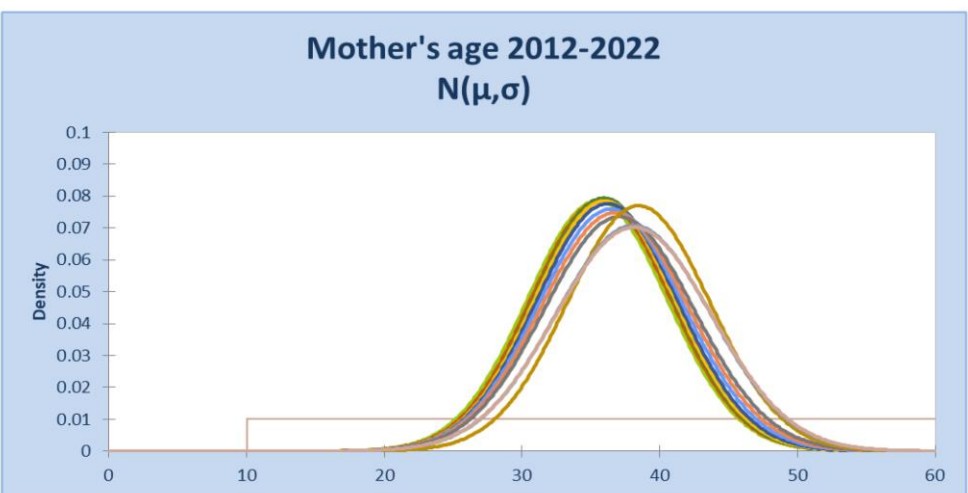

**Figure 6.** Maternal age at application. Normal distributions per year. Source: data taken from EETAA (2011–2022).

Methods of analysis. The statistical method of ANCOVA, which can combine ANOVA and regression for adjustment of the linear effect of the covariate, thus providing a clearer picture, was used. Preliminary ANOVA tests were conducted for MINC and FINC for all samples. Factorial ANCOVA is conducted with more than one independent variable. The model includes two covariates. The categorical variables included in the model here (factors) are mother's nationality, mother's education, mother's family status, mother's work status, region of residence. We controlled by design the external validity by filtering the age range out from all samples, for the sake of a more homogeneous sample. The metric variables are father's income and mother's age measured in years, as covariates of the dependent variable, which is a also a continuous variable, mother's income, while father's income is regarded as both dependent and covariate, as mentioned above. Instead of running a MANCOVA model, we chose to examine the two separately. The total

family income that is registered on the application is not used either as dependent or explanatory, because MINC and FINC data offer much better insight. Group membership was determined randomly [31]. In the linear model, interaction terms were also added.

In order to exploit the cross-section explanatory power of the data, covariance analysis is utilized, for it reduces all the unwanted predictable variance with the covariates and enables interactions among categorical and quantitative variables. The relationship tested provided a weak explanation for datasets before the outburst of the pandemic with MINC as the dependent variable, and a more satisfying level with FINC as the dependent variable. As for the period closer to or after the COVID-19 pandemic outburst, statistics offered a very different perspective. To increase statistical power and ensure an accurate measurement of the true relationship between the dependent variable and the independent variable by reducing the mean square error, we keep all covariates in the sample as control variables, unless they are made statistically redundant, in which case they are to be removed. This study recognizes the potential benefits of reduced error variance to increase statistical power and improve precision, as well as the importance of running preliminary ANOVA tests that allow for comparison.

PCA Analysis. Before the regression, a PCA analysis was conducted to better describe the data and gain insight from the relationships revealed between variables, understand how each variable is associated with one another, as did, e.g., nationality, even though almost a constant, and employment, which was however differently categorized in the pre- and post-pandemic group.

The length of the vector made it quite clear that the most explanatory variables would be region, mother's income and mother's age. Still, as shown in the PCA, the two axes chosen cannot fully grasp the phenomenon, but only at a rather moderate level, 40.87% and 55.26%, respectively.

PCA analysis (Figure 7) was expected to assist in choosing the dimensions of the data that add life fulfilment and eliminate the ones that do not. Education, with so many missing values, was easily left out.

**(a)** **(b)**

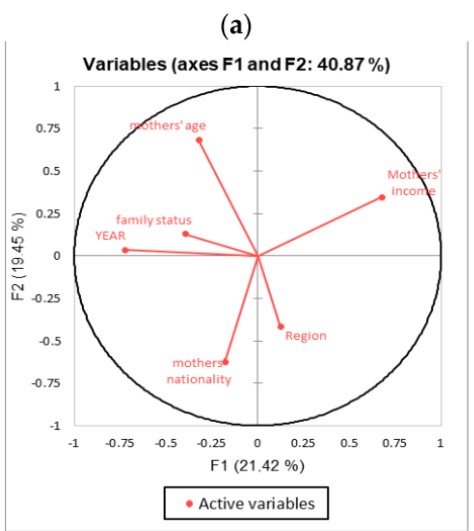 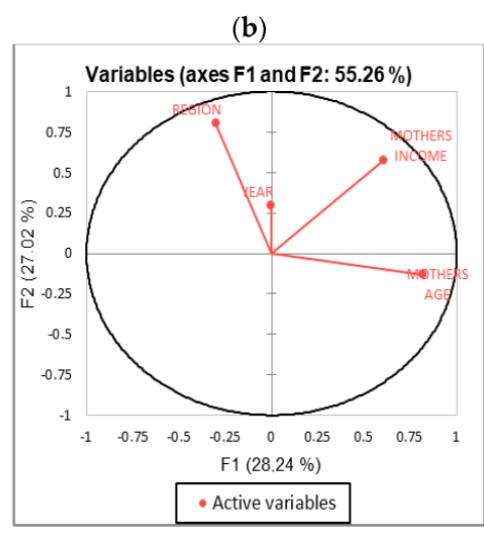

**Figure 7.** PCA analysis (**a**) of the 6 variables; (**b**) of the 4 variables.

We applied ANCOVA on each subsample of each year (2012–2022) assuming a linear relationship between the dependent variable and the predictor variables (covariates). The model for MINC without the interactions is the following:

$$MINC_i = \beta_0 + b_1 FINC_{ij} + b_2 MAGE_{ij} + b_3 MNAT_{ij} + b_4 MWORK_{ij} + b_5 MFAM_{ij} + b_6 MED_{ij} + b_7 REG_{ij} + \epsilon_i \quad (1)$$

The FINC model for each dataset is:

$$FINC_i = \beta_0 + b_1 MINC_{ij} + b_2 MAGE_{ij} + b_3 MNAT_{ij} + b_4 MWORK_{ij} + b_5 MFAM_{ij} + b_6 MED_{ij} + b_7 REG_{ij} + \epsilon_i \quad (2)$$

The model was run twice in different forms with the interaction's terms included for each dataset:

$$MINC_i = \beta_0 + b_1 FINC_{ij} + b_2 MAGE_{ij} + b_3 MNAT_{ij} + b_4 MWORK_{ij} + b_5 MFAM_{ij} + b_6 MED_{ij} + b_7 REG_{ij}$$
$$+ b_{2\_7} MAGE * REG_{IJ} + b_{2\_3} MAGE * MNAT_{ij} + b_{2\_5} MAGE * MFAM_{ij}$$
$$+ b_{2\_4} MAGE * MWORK_{ij} + \epsilon_i$$

### 3. Findings and Discussion

Tables 2 and 3, presents the estimated coefficients of the models. The statistically significant coefficients are marked in bold.

**Table 2.** Estimated parameters.

| Source | Value | Standard Error | t | Pr > \|t\| | Lowerbound (95%) | Upperbound (95%) |
|---|---|---|---|---|---|---|
| Intercept | 7031.152 | 305.675 | 23.002 | <0.0001 | 6432.025 | 7630.279 |
| MOTHER'S AGE | 184.467 | 7.974 | 23.133 | <0.0001 | 168.837 | 200.096 |
| FATHER'S INCOME | −0.067 | 0.007 | −9.752 | <0.0001 | −0.080 | −0.053 |
| REGION-1 | 0.000 | 0.000 | | | | |
| REGION-2 | −1576.623 | 160.212 | −9.841 | <0.0001 | −1890.642 | −1262.605 |
| REGION-3 | −578.660 | 275.156 | −2.103 | 0.035 | −1117.970 | −39.350 |
| REGION-4 | −681.610 | 249.265 | −2.734 | 0.006 | −1170.172 | −193.048 |
| REGION-5 | −871.240 | 196.573 | −4.432 | <0.0001 | −1256.527 | −485.954 |
| REGION-6 | −2974.108 | 407.089 | −7.306 | <0.0001 | −3772.009 | −2176.207 |
| REGION-7 | −1457.506 | 237.992 | −6.124 | <0.0001 | −1923.974 | −991.038 |
| REGION-8 | −426.700 | 276.340 | −1.544 | 0.123 | −968.330 | 114.930 |
| REGION-9 | 187.660 | 172.024 | 1.091 | 0.275 | −149.510 | 524.829 |
| REGION-10 | −677.235 | 227.224 | −2.980 | 0.003 | −1122.598 | −231.872 |
| REGION-11 | 72.978 | 376.387 | 0.194 | 0.846 | −664.746 | 810.702 |
| REGION-12 | −466.950 | 243.854 | −1.915 | 0.056 | −944.908 | 11.008 |
| REGION-13 | −1635.383 | 183.879 | −8.894 | <0.0001 | −1995.788 | −1274.979 |
| WORK STATUS-1 | 0.000 | 0.000 | | | | |
| WORK STATUS-2 | 399.782 | 93.324 | 4.284 | <0.0001 | 216.866 | 582.698 |
| WORK STATUS-3 | −653.705 | 135.492 | −4.825 | <0.0001 | −919.272 | −388.139 |
| WORK STATUS-4 | −2194.524 | 145.098 | −15.124 | <0.0001 | −2478.919 | −1910.129 |
| NATIONALITY-1 | 0.000 | 0.000 | | | | |
| NATIONALITY-2 | −2237.517 | 289.911 | −7.718 | <0.0001 | −2805.746 | −1669.288 |
| NATIONALITY-3 | −3629.654 | 159.815 | −22.712 | <0.0001 | −3942.893 | −3316.415 |
| FAMILY STATUS-1 | 0.000 | 0.000 | | | | |
| FAMILY STATUS-2 | −6291.432 | 226.128 | −27.822 | <0.0001 | −6734.646 | −5848.217 |
| FAMILY STATUS-3 | −4057.005 | 195.614 | −20.740 | <0.0001 | −4440.411 | −3673.600 |
| FAMILY STATUS-4 | −2883.792 | 557.334 | −5.174 | <0.0001 | −3976.175 | −1791.409 |

**Table 3.** ANCOVA Interaction Results. 2012–2022 (We thank a reviewer for her/his suggestion to include Table 3).

| Academic Year | $R^2$ | Interactions |
|---|---|---|
| 2012 | 5% | Family × work |
| 2013 | 6% | Family × work |
| 2014 | 6% | Family × work |
| 2015 | 6% | Family × work |
| 2016 | 6% | Family × work |
| 2017 | 6% | Family × work |
| 2018 | 2% | Age × work |
| 2019 | 5% | Work |
| 2020 | 32% | Age × work |
| 2021 | 27% | Age × work |
| 2022 | 32% | Age × work |

As to the interactions, in the pre COVID-19 pandemic period, the variable "Family*Work status" was the most influential in the ANCOVA. A light change in the trend appears in the school year 2018 (2017–2018) in Table 3. This transformation continues in the next school year 2019 (2018–2019) prior to the COVID-19 pandemic. A new trend started in the school year 2020 (2019–2020), with variable "Mother's age* Mother's work status" in the interactions as the most influential on the response variable, "Mother's income".

After the outbreak of COVID-19, this trend goes on in the 2021 (2020–2021) and 2022 (2021–2022) school years and, by extension, "Mother's age* work status" is the variable that affects a mother's decision to apply for daycare. The COVID-19 pandemic emphasized the importance of the role Mother's age plays in both her fertility and her position in the labour market.

### 3.1. Calibration of the 2012 Sample

Performing an ANOVA test to the data set, we can reject the null hypothesis that there is no effect of the explanatory variables (Table 4).

**Table 4.** Analysis of variance: Fisher's test.

| Source | DF | Sum of Squares | Mean Squares | F | Pr > F | *p*-Values Signification Codes |
|---|---|---|---|---|---|---|
| Model | 22 | $1.60 \times 10^{11}$ | $7.28 \times 10^{9}$ | 109.958 | <0.0001 | *** |
| Error | 47,536 | $3.15 \times 10^{12}$ | $6.62 \times 10^{7}$ | | | |
| Corrected Total | 47,558 | $3.31 \times 10^{12}$ | | | | |

Computed against model Y = Mean(Y); Signification codes: *** $p < 0.001$.

Therefore, we can conclude with confidence that the chosen variables bring a significant amount of information to our analysis. Then, we check the information power of the explanatory variables, by examining the Type I SS and Type III SS tables (see Tables 5 and 6). The Type I SS table is constructed by adding variables to the model one by one, and by evaluating the impact of each on the model sum of squares (Model SS). In consequence, in Type I SS, the order in which the variables are selected will influence the results. The lower the F probability corresponding to a given variable, the stronger the impact of the variable on the model, as it is before the variable is added to it. We notice that the chosen variables all bring power to the model, and are all important.

**Table 5.** Type I Sum of Squares analysis (Mother's income).

| Source | DF | Sum of Squares | Mean Squares | F | Pr > F |
|---|---|---|---|---|---|
| Father's Inc | 1 | $1.015 \times 10^{9}$ | $1.015 \times 10^{9}$ | 15.339 | <0.0001 |
| Mother's Age | 1 | $3.236 \times 10^{10}$ | $3.236 \times 10^{10}$ | 489.016 | <0.0001 |
| Region | 12 | $2.666 \times 10^{10}$ | $2.221 \times 10^{9}$ | 33.565 | <0.0001 |
| Work Status | 3 | $1.850 \times 10^{10}$ | $6.167 \times 10^{9}$ | 93.177 | <0.0001 |
| Nationality | 2 | $2.429 \times 10^{10}$ | $1.215 \times 10^{10}$ | 183.526 | <0.0001 |
| Family Status | 3 | $5.727 \times 10^{10}$ | $1.909 \times 10^{10}$ | 288.457 | <0.0001 |

**Table 6.** Type II Sum of Squares analysis (Mother's income).

| Source | DF | Sum of Squares | Mean Squares | F | Pr > F |
|---|---|---|---|---|---|
| Mother's Age | 1 | $3.660 \times 10^{10}$ | $3.660 \times 10^{10}$ | 552.959 | <0.0001 |
| Father's Inc. | 1 | $6.419 \times 10^{9}$ | $6.419 \times 10^{9}$ | 96.986 | <0.0001 |
| Region | 12 | $2.666 \times 10^{10}$ | $2.222 \times 10^{9}$ | 33.571 | <0.0001 |
| Work Status | 3 | $2.252 \times 10^{10}$ | $7.505 \times 10^{9}$ | 113.403 | <0.0001 |
| Nationality | 2 | $2.303 \times 10^{10}$ | $1.151 \times 10^{10}$ | 173.974 | <0.0001 |
| Family Status | 3 | $5.727 \times 10^{10}$ | $1.909 \times 10^{10}$ | 288.457 | <0.0001 |

We can see that the *p*-value for Region-11 (islands in North Aegean) parameter is 0.846 and for Region-9 (Attica) parameter 0.275, and that the corresponding confidence range includes 0. This confirms the weak impact of these two regions on the model. If we look at the parameter corresponding to Regions 9 and 11, it seems that, for a given age and profession, coming from these two areas means a small increase in the mother's income.

The chart in Figure 8 shows the predicted values versus the observed values. The Confidence intervals allow identification of potential outliers. No suspicious residuals were detected that could lead to rejecting the hypothesis of normality, and there were no potential outliers.

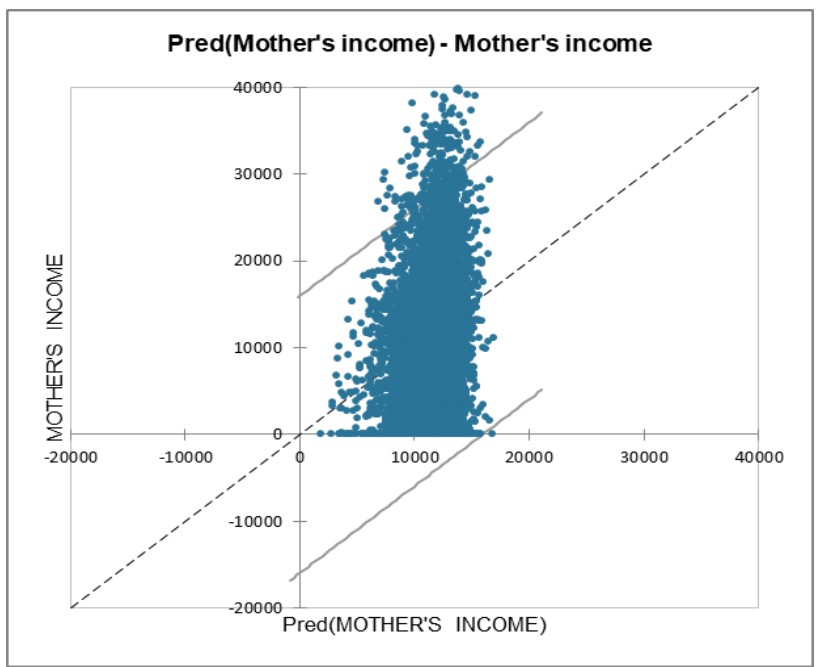

**Figure 8.** Predicted versus observed values on the dependent variable.

Given the *p*-value of the F statistic computed in the ANOVA table, and given the significance level of 5%, the information brought by the explanatory variables is significantly better than that which a basic mean would bring. Based on the Type III sum of squares, the following variables bring significant information to explain the variability of the dependent variable Mother's income: Husband's income, Age, Region, Profession and Family status. Among the explanatory variables, based on the Type III sum of squares, the variable FAMILY STATUS is the most influential.

### 3.2. Calibration of the 2013–2019 Data Samples

In the academic years following 2011–2012, the pattern between the dependent variable and the most explanatory variable remained. Table 7 gives a short review of the results.

**Table 7.** 2013–2019: The explanatory variable remains the same.

| Academic Year | $R^2$ | Explanatory Variable |
|---|---|---|
| 2013 | 6% | Family status |
| 2014 | 5% | Family status |
| 2015 | 5% | Family status |
| 2016 | 5% | Family status |
| 2017 | 5% | Family status |
| 2018 | 5% | Family status |
| 2019 | 5% | Family status |

### 3.3. Calibration of the 2020–2022 Data

In the last three years, regressions show that the model fits the data at a much more satisfying level, while at the same time it offers insight. While all the previous years seemed to give somewhat inexplicable repeated results, the 2020–2022 report explains the pre-COVID period, but at the same time the post-recession and the post COVID-19 years examined here. Table 8 presents the obtained results.

**Table 8.** 2020–2022 Results.

| Academic Year | $R^2$ | Explanatory Variable |
|---|---|---|
| 2020 | 30% | Work status |
| 2021 | 26% | Mother's age [1] |
| 2022 | 31% | Mother's age |

[1] The predictive power of ANCOVA appears strengthened.

In the 2020 sample (Table 9), among the explanatory variables work status of the mother is the most influential, while based on the Type III sum of squares, the variable Age × work status is the most influential in the ANCOVA interactions.

**Table 9.** Interactions ANCOVA2020–2022.

| Academic Year | $R^2$ | Interactions |
|---|---|---|
| 2020 | 32% | Age × work status |
| 2021 | 27% | Age × work status |
| 2022 | 32% | Age × work status |

In the 2021 sample, among the explanatory variables, based on the Type III sum of squares, the variable Mothers' AGE is the most influential, while based on the Type III sum of squares, the variable mother's Age × work status is also the most influential in the ANCOVA interactions.

In the 2022 sample, given the R2, 31% of the variability of the dependent variable MOTHERS INCOME is explained by the six explanatory variables. Among the explanatory variables, based on the Type III sum of squares, the variable Mother's AGE is the most influential, while based on the Type III sum of squares, the variable Mother's Age × work status is also the most influential in the ANCOVA interactions.

It is important to examine the results of the analysis of variance tables for all samples, as it enables us to determine whether or not the explanatory variables bring significant information (null hypothesis H0) to the model. In other words, it is a way of asking whether it is valid to use the mean to describe the whole population, or whether the information brought by the explanatory variables is of value or not. For the 2020–2022 samples, the tables below justify the choice of variables (Tables 10–12).

**Table 10.** 2020 sample. Analysis of variance (MOTHERS INCOME). (a) 6 variables; (b) 11 variables.

| Source | DF | Sum of Squares | Mean Squares | F | Pr > F |
|---|---|---|---|---|---|
| Model | 22 | $5.223 \times 10^{11}$ | $2.374 \times 10^{10}$ | 1486.786 | <0.0001 |
| Error | 75,286 | $1.202 \times 10^{12}$ | $1.597 \times 10^7$ | | |
| Corrected Total | 75,308 | $1.724 \times 10^{12}$ | | | |

| Source | DF | Sum of Squares | Mean Squares | F | Pr > F |
|---|---|---|---|---|---|
| Model | 79 | $5450 \times 10^{11}$ | $6.899 \times 10^9$ | 440.011 | <0.0001 |
| Error | 75,229 | $1.179 \times 10^{12}$ | $1.568 \times 10^7$ | | |
| Corrected Total | 75,308 | $1.724 \times 10^{12}$ | | | |

**Table 11.** 2021 sample. Analysis of variance (MOTHERS INCOME). (a) 6 variables; (b) 10 variables.

| Source | DF | Sum of Squares | Mean Squares | F | Pr > F |
|---|---|---|---|---|---|
| Model | 24 | $1.210 \times 10^{12}$ | $5.043 \times 10^{10}$ | 2392.398 | <0.0001 |
| Error | 162,982 | $3.435 \times 10^{12}$ | $2.108 \times 10^{7}$ | | |
| Corrected Total | 163,006 | $4.646 \times 10^{12}$ | | | |
| **Source** | **DF** | **Sum of Squares** | **Mean Squares** | **F** | **Pr > F** |
| Model | 46 | $1.267 \times 10^{12}$ | $2.755 \times 10^{10}$ | 1328.678 | <0.0001 |
| Error | 162,960 | $3.379 \times 10^{12}$ | $2.073 \times 10^{7}$ | | |
| Corrected Total | 163,006 | $4.646 \times 10^{12}$ | | | |

**Table 12.** 2022 sample. Analysis of variance (MOTHERS INCOME). (a) 11 variables; (b) 6 variables.

| Source | DF | Sum of Squares | Mean Squares | F | Pr > F |
|---|---|---|---|---|---|
| Model | 49 | $1.801 \times 10^{12}$ | $3.644 \times 10^{10}$ | 1640.939 | <0.0001 |
| Error | 173,236 | $3.326 \times 10^{12}$ | $1.929 \times 10^{7}$ | | |
| Corrected Total | 173,285 | $5.039 \times 10^{12}$ | | | |
| **Source** | **DF** | **Sum of Squares** | **Mean Squares** | **F** | **Pr > F** |
| Model | 25 | $1.751 \times 10^{12}$ | $7.005 \times 10^{10}$ | 3086.823 | <0.0001 |
| Error | 173,260 | $3.932 \times 10^{12}$ | $2.269 \times 10^{7}$ | | |
| Corrected Total | 173,285 | $5.683 \times 10^{12}$ | | | |

Covariance provides insight into how two variables are related. It refers to the measure of how two random variables in a data set will change together. A positive covariance means that the two variables at hand are positively related, and they move in the same direction.

Robustness checks: Assuming that the procedure is sufficiently robust, the failure to adequately account for the assumptions will result in only a small and insignificant deviation from the actual result, had the assumptions been met. Having created boxplots for each group, we have seen that the values in each group are roughly equal. (Not all outputs are shown in the text due to space limitations. If needed, please request from the authors). An ANCOVA preliminary analysis testing the assumptions of homogeneity of variance and the regression slopes, aka the interaction between the factors and the covariates, is followed to make sure the assumptions are all met. In summary, since ANCOVA is about adjusting group means as if people reacted equally on the covariates, the utility of each covariate will be analyzed. When the influence of the covariate on the dependent variable is important, the aforementioned group adjustments provide lower F-values and *p*-values.

Levene's (1960) test is applied as appropriate to test for homogeneity of variance. When it is not significant, then ANCOVA can be conducted [32]. Levene's test, though, might be overstating the problem in this case,, where samples sizes are large and/or unequal providing a smaller *p*-value ($p < 0.05$), while understating it in small samples. In this case, where the samples are large, the analysis could disregard any violation of the assumptions and continue as if assumptions were indeed satisfied [33].

Seeking to determine the effect of nationality, work status, family status and region of residence on the applicant's income (MINC & FINC) while controlling for mother's age, mother's income and father's income alternately, data was first screened for missing data and outliers and later tested on the assumptions. Data screening led to the transformation of MINC FINC, MAGE in order to eliminate all cases with income and ages above the cutoff point. Linearity of the covariate MAGE was not assessed, as mother's age is categorical.

The goal is to measure the effect of each dependent variable and their contribution to the differences in the independent variables. In this study, in order to remove the

effects of all other dependent variables (i.e., father's income), it was chosen to treat them as covariates.

ANCOVA results (see Tables 13 and 14) indicate significant main effects for the most influential independent variable in each yearly sample. The power between the dependent and the most explanatory variable in the pre-COVID-19 period remained low for MINC. In the concluding years, the linear model fits the data at a much more satisfying level for MINC as the dependent variable, offering insight. The years of the program before the COVID-19 outburst had quite similar results, while 2019–2022 reported a change.

Even though $R^2$ for MINC regression remained low until the outburst of COVID-19, 5% of its variability explained, on average, the results reported in the preliminary ANOVA and the ANCOVA table each year followed almost the same pattern with a FINC regression that had a much better fit with the data, an average of 30%. The analysis of covariance was conducted for both MINC and FINC as the two variables of interest forming the family income. FINC (2013 sample) was also removed from the MINC analysis because of the appearance of a multicollinearity effect.

**Table 13.** Preliminary MINC & FINC ANOVA Analysis Results: 2012–2022.

| | | | | SS III Analysis MINC ANOVA | | | | |
|---|---|---|---|---|---|---|---|---|
| Year | Variable | DF | SS | MS | F | Pr > F | *p*-Values Signification | $R^2$ |
| 2012 | MFAM | 3 | $5.100 \times 10^{10}$ | $1.700 \times 10^{10}$ | 253.698 | <0.0001 | *** | 0.04 |
| 2013 | MFAM | 3 | $8.689 \times 10^{10}$ | $2.896 \times 10^{10}$ | 428.373 | <0.0001 | *** | 0.04 |
| 2014 | MFAM | 3 | $6.754 \times 10^{10}$ | $2.251 \times 10^{10}$ | 388.220 | <0.0001 | *** | 0.04 |
| 2015 | MFAM | 3 | $6.465 \times 10^{10}$ | $2.155 \times 10^{10}$ | 443.149 | <0.0001 | *** | 0.03 |
| 2016 | MFAM | 3 | $7.067 \times 10^{10}$ | $2.356 \times 10^{10}$ | 502.476 | <0.0001 | *** | 0.04 |
| 2017 | MFAM | 3 | $1.443 \times 10^{10}$ | $4.810 \times 10^{9}$ | 104.669 | <0.0001 | *** | 0.04 |
| 2018 | MFAM | 3 | $1.125 \times 10^{11}$ | $3.751 \times 10^{10}$ | 830.266 | <0.0001 | *** | 0.04 |
| 2019 | MFAM | 3 | $1.068 \times 10^{11}$ | $3.561 \times 10^{10}$ | 826.866 | <0.0001 | *** | 0.04 |
| 2020 | MWORK | 3 | $4.353 \times 10^{11}$ | $1.451 \times 10^{11}$ | 8934.420 | <0.0001 | *** | 0.29 |
| 2021 | REGION | 12 | $3.304 \times 10^{10}$ | $2.754 \times 10^{9}$ | 138.273 | <0.0001 | *** | 0.26 |
| 2022 | REGION | 12 | $5.528 \times 10^{10}$ | $4.607 \times 10^{9}$ | 236.010 | <0.0001 | *** | 0.332 |
| | | | | SS III Analysis FINC ANOVA | | | | |
| Year | Variable | DF | SS | MS | F | Pr > F | *p*-Values Signification | $R^2$ |
| 2012 | MWORK | 3 | $2.254 \times 10^{11}$ | $7.514 \times 10^{10}$ | 2452.660 | <0.0001 | *** | 0.18 |
| 2013 | MFAM | 3 | $8.129 \times 10^{10}$ | $2.710 \times 10^{10}$ | 924.278 | <0.0001 | *** | 0.23 |
| 2014 | MFAM | 3 | $6.773 \times 10^{10}$ | $2.258 \times 10^{10}$ | 921.941 | <0.0001 | *** | 0.23 |
| 2015 | MFAM | 3 | $9.184 \times 10^{10}$ | $3.061 \times 10^{10}$ | 1643.466 | <0.0001 | *** | 0.31 |
| 2016 | MFAM | 3 | $1.115 \times 10^{11}$ | $3.717 \times 10^{10}$ | 2127.524 | <0.0001 | *** | 0.29 |
| 2017 | MFAM | 3 | $1.330 \times 10^{11}$ | $4.432 \times 10^{10}$ | 2898.021 | <0.0001 | *** | 0.35 |
| 2018 | MFAM | 3 | $1.445 \times 10^{11}$ | $4.815 \times 10^{10}$ | 3278.915 | <0.0001 | *** | 0.33 |
| 2019 | MWORK | 3 | $5.766 \times 10^{11}$ | $1.922 \times 10^{11}$ | 12,777.390 | <0.0001 | *** | 0.3 |
| 2020 | MFAM | 3 | $5.165 \times 10^{11}$ | $1.722 \times 10^{11}$ | 4443.23 | <0.0001 | *** | 0.66 |
| 2021 | MFAM | 3 | $1.238 \times 10^{12}$ | $4.126 \times 10^{11}$ | 9303.423 | <0.0001 | *** | 0.159 |
| 2022 | REGION | 12 | $9.396 \times 10^{10}$ | $7.830 \times 10^{9}$ | 178.942 | <0.0001 | *** | 0.2 |

Signification codes: *** $p < 0.001$.

**Table 14.** MINC & FINC ANCOVA Analysis Results. 2012–2022.

| Year | Variable | DF | SS | MS | F | Pr > F | *p*-Values Signification | $R^2$ |
|---|---|---|---|---|---|---|---|---|
| | | | **SS III MINC ANCOVA** | | | | | |
| 2012 | MFAM | 3 | $5.727 \times 10^{10}$ | $1.909 \times 10^{10}$ | 288.457 | <0.0001 | *** | 0.05 |
| 2013 | MFAM | 3 | $1.081 \times 10^{11}$ | $3.603 \times 10^{10}$ | 542.481 | <0.0001 | *** | 0.06 |
| 2014 | MFAM | 3 | $8.478 \times 10^{10}$ | $2.826 \times 10^{10}$ | 493.978 | <0.0001 | *** | 0.05 |
| 2015 | MFAM | 3 | $8.765 \times 10^{10}$ | $2.922 \times 10^{10}$ | 609.983 | <0.0001 | *** | 0.05 |
| 2016 | MFAM | 3 | $9.568 \times 10^{10}$ | $3.189 \times 10^{10}$ | 689.670 | <0.0001 | *** | 0.05 |
| 2017 | MFAM | 3 | $1.169 \times 10^{11}$ | $3.898 \times 10^{10}$ | 859.676 | <0.0001 | *** | 0.05 |
| 2018 | MFAM | 3 | $1.309 \times 10^{11}$ | $4.362 \times 10^{10}$ | 973.033 | <0.0001 | *** | 0.05 |
| 2019 | MFAM | 3 | $1.354 \times 10^{11}$ | $4.515 \times 10^{10}$ | 1060.455 | <0.0001 | *** | 0.05 |
| 2020 | MWORK | 3 | $4.303 \times 10^{11}$ | $1.434 \times 10^{11}$ | 8982.707 | <0.0001 | *** | 0.303 |
| 2021 | MAGE | 1 | $4.904 \times 10^{10}$ | $4.904 \times 10^{10}$ | 2500.473 | <0.0001 | *** | 0.26 |
| 2022 | MAGE | 1 | $3.814 \times 10^{10}$ | $3.814 \times 10^{10}$ | 1977.162 | <0.0001 | *** | 0.34 |
| | | | **SS III FINC ANCOVA** | | | | | |
| 2012 | MWORK | 3 | $2.236 \times 10^{11}$ | $7.455 \times 10^{10}$ | 2479.779 | <0.0001 | *** | 0.19 |
| 2013 | MAGE | 1 | $4.767 \times 10^{10}$ | $4.767 \times 10^{10}$ | 1670.906 | <0.0001 | *** | 0.25 |
| 2014 | MFAM | 3 | $7.612 \times 10^{10}$ | $2.537 \times 10^{10}$ | 1060.219 | <0.0001 | *** | 0.24 |
| 2015 | MAGE | 1 | $2.809 \times 10^{10}$ | $2.809 \times 10^{10}$ | 1544.750 | <0.0001 | *** | 0.31 |
| 2016 | MFAM | 3 | $1.229 \times 10^{11}$ | $4.097 \times 10^{10}$ | 2396.672 | <0.0001 | *** | 0.31 |
| 2017 | MFAM | 3 | $1.454 \times 10^{11}$ | $4.846 \times 10^{10}$ | 3232.650 | <0.0001 | *** | 0.36 |
| 2018 | MFAM | 3 | $1.525 \times 10^{11}$ | $5.084 \times 10^{10}$ | 3503.464 | <0.0001 | *** | 0.34 |
| 2019 | MWORK | 3 | $5.731 \times 10^{11}$ | $1.910 \times 10^{11}$ | 12,899.156 | <0.0001 | *** | 0.31 |
| 2020 | MFAM | 3 | $4.941 \times 10^{11}$ | $1.647 \times 10^{11}$ | 4284.186 | <0.0001 | *** | 0.172 |
| 2021 | MFAM | 3 | $1.240 \times 10^{12}$ | $4.134 \times 10^{11}$ | 9347.308 | <0.0001 | *** | 0.16 |
| 2022 | REGION | 12 | $9.133 \times 10^{10}$ | $7.611 \times 10^{9}$ | 174.436 | <0.0001 | *** | 0.2 |

Signification codes: *** $p < 0.001$.

ANCOVA results indicate a significant main effect for almost all variables in the model with very few exceptions, i.e., in the 2021 sample, MINC was made redundant and was removed from the FINC analysis of covariance, and FINC was made redundant from the MINC analysis of covariance and was removed for that same year. ANCOVA results for FINC revealed a significant main effect for work status in the 2012 sample, for mother's age in the 2013 sample and in the 2015 sample, for in the 2022 sample, and a significant main effect for family status in all the other samples spanning 2012–2022. Results for MINC indicated a significant main effect for family status for the 2012 sample and remained significant throughout all years until 2019. A significant main effect in 2019 for MWORK is also shown. The covariate of mother's age significantly influenced the dependent variable of Mother's income in 2021 and 2022. It should also be mentioned that all variables demonstrated significant main effects, while the contrary applies for their interaction terms. The goodness of fit for the two variables under examination was reversed, with MINC reaching $R^2 = 34\%$ in the 2022 sample.

In summary, the results from the covariance analysis show that family status exhibited significant main effects year after year on the dependent variable of the univariate analysis,

on both MINC and FINC, prior to the COVID-19 pandemic outburst. In 2020 (school year 2019–2020), ANCOVA results indicate MWORK "work status" as the most influential variable on the dependent MINC. In the post-COVID 19 pandemic period, MAGE "Mother's age" has been indicated as the most influential on the dependent MINC variable. Region was indicated for the first time as influential on the dependent FINC in the post COVID-19 pandemic period in the 2022 analysis. Region is also indicated as the most influential on the DV FINC and MINC in the preliminary ANOVA analysis of the 2022 sample.

Drawing upon the concluding results, a more inclusive institutional framework seems to be dictated as the precondition of the viability of kindergartens in the future. As the family itself will evolve as an institution in a more diverse way, what future society will thank us for is making room for change. Older generations having children is one sign, refugee children another, each unique and urgent at the same time. The authors of [34] present an historic opportunity to place the protection of children at the heart of the policy actions of every nation and to build a world where all children everywhere enjoy freedom from fear and from violence in all its forms. Violence against children knows no boundaries. It takes place in institutions designed for their care and protection, and also within the home. Although of epidemic proportions, violence against children often remains hidden and socially condoned. Promoting inclusiveness in education, and allowing for a decent economic growth, within the aforementioned framework, will bring us closer to reduced inequality both within and among countries, within and among families, within and among genders.

## 4. Conclusions

In this paper we examined the relationship between the main driver of the daycare subsidized program, which is family income, and the characteristics of all individual mothers. In our hypotheses, we came to believe that all the elements of a mother's profile have played a significant role in the evolution of the program.

In summary, based on the findings of our analysis, the following variables are significant in explaining Father's Income/Mother's Income: region, mother's age, mother's nationality, mother's family and work status (salary), with region being the most influential.

The results from the linear model show that family has been a priority for the 2012–2019 cohorts participating in the program prior to the COVID-19 pandemic. The situation started changing in the 2020 school year (2019–2020), with ANCOVA results showing "work" as the influential variable, and became very different in the post-COVID 19 years where the variable "Mother's age" is the most influential on the "Mother's income" variable.

According to the results, mother's family status was the main asset behind her decision to apply before the COVID-19 outbreak. However, for the post-COVID group, the main factor affecting the decision is the mother's advancing age, indicating the greater level of uncertainty experienced by women in their reproductive years in times of crisis.

Our findings reveal that the outbreak of the COVID-19 pandemic acted as an accelerator of long-term trends, such as the increase in inequalities, but also as fertile ground for the undermining of labor and other social rights and the erosion of democratic institutions. They also emphasize that, during and after-COVID-19, data and trends of asymmetric recovery have begun to be clearly erased, with consolidation of imbalances and worsening of inequalities. This is in concurrence with recent research [35].

By growing the already given clusters in a region and promoting policies that create a sense of place, regions will be made more competitive and appealing to firms and workers [36]. Cities, people and families can grow in every region where the chance for social engagement is abundant, caregivers, caretakers and caterers included. Supporting the concurrent belief of [37]: "The most valuable of all capital is that invested in human beings; and of that capital the most precious part is the result of the care and influence of the mother".

However, our study is limited by the data available, and the lack of consideration of other events, such as the issue of Ukraine–Russia, increasing inflation, and the increase

of prices of goods and commodities. It is expected that these events might have different effects across regions. In the future, it will be interesting to see how all these events interfere with our variables and to test if there will be any major regional differences across the variables presented here.

**Author Contributions:** Writing—original draft, C.S. and S.A.K. All authors have read and agreed to the published version of the manuscript.

**Funding:** This research received no external funding.

**Conflicts of Interest:** The authors declare no conflict of interest.

## Appendix A

**Table A1.** The 13 Administrative Regions in Greece.

| Region | Area |
| --- | --- |
| 1 | East Macedonia and Thrace |
| 2 | Central Macedonia |
| 3 | West Macedonia |
| 4 | Epirus |
| 5 | Thessaly |
| 6 | Ionian islands |
| 7 | West Greece |
| 8 | Central Greece |
| 9 | Attica |
| 10 | Peloponnese |
| 11 | North Aegean |
| 12 | South Aegean |
| 13 | Crete |

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
