# Peer review of "Female Labor Participation and Mother’s Daycare in Greece Pre- and Post-COVID-19 Pandemic: A Regional Analysis"

_sustainability, doi:10.3390/su151410920_

Round 1

Reviewer 1 Report

see my comments as attached

Author Response

According to the suggestions and the comments of Reviewer 1 we made the requested amendments

Reviewer 2 Report

Female labor participation and mother’s daycare in Greece pre and post-covid19 pandemic: A regional analysis were explored by the author(s). The title is relevant to the scope of Sustainability, but the manuscript requires significant improvement to exhibit quality and capture immense readability. Therefore, some observations are indicated below:

Abstract

The abstract is well presented by the authors. However, the authors failed to briefly state the contributions and offer one or two recommendations in the abstract.

1.     Introduction

I suggest you rewrite the introductory section to show a clear line of thought. In the present form, the introduction is structurally deficient.

Comment: To a large extent, the Covid-19 pandemic is no more a catalyst in daily activities in the last 12 months or more. However, the claim of the authors in page 1, lines 21-22 is alarming. Therefore, it would be best if the authors could provide some most statistics of the effects of Covid-19 on decision making of governments and private organizations in Greece.

UN is not defined at the first point of usage which makes understanding the lines very difficult. Also, check EU, UNICEF etc.

Figures 2, 3 and 4 are blurred and too small. Please amend them.

I could not find the research question and objective in the manuscript.

Change ‘’doesn’t’’ in line 108 to ‘’does not’’

The assertion made by the authors in lines 113-114 is not enough as the significance of the study or the gap in the literature. The authors failed to make a solid justification for the uniqueness of the study and no single contributions were suggested. This, in my opinion, render the manuscript not different from exiting studies.

2.     Materials and Methods

What is the purpose of Figure 5? It is blurred and small. Please amend it.

The statistics in lines 114-115 seem not clear. You mentioned in the previous section that application to the surge after covid-19 pandemic. In line 114, you claimed that the program received 827, 558 applications during pre-covid-19 between 2011 and 2020. It would be best if you separate the data (applications) into pre-covid19 and during covid-19?

The authors wrote 36 y.o. and 38 y.o. in lines 193 and 194. Please write the y.o. in full form

Please identify the theory guiding your methods—especially the panel data approach.

Also, the authors need to discuss the methodological gap.

3.     Results and discussion

First, the authors was busy explaining the method in the two first paragraphs which comprises of lines 239-252. It would be best if you move the explanation to section 2.

Second, from the title of the manuscript, the authors proposed a regional analysis of Female labor participation and mother’s daycare pre and post-covid19 pandemic in Greece. However, in the results and discussion section, regional results were missing and no analysis pertaining to regions in Greece was found.

Third, merely describing results means the authors perhaps lack sufficient information for analysis. I suggest that the authors validate the results' interpreted with earlier arguments.

I expect the authors to provide results mainly focused on pre-covid, during covid-19 and post covid-19. Then, ensure you compare these results and delve into the key factors affecting

4.     Conclusion

The conclusion is well written but too short, and policy recommendations needs improvement.

I recommend the acceptance of the manuscript upon major corrections.

Moderate

Author Response

According to the comments and suggestions of Reviewer 2 we made the necessary amendments.

Reviewer 3 Report

The article makes a quantitive analysis about Greek mothers applying for childcare in a European program aimed t promote gender equality. The article has the ambition to discuss the COVID pandemic impact on women's work. Data concern more than one million mothers. However the interpretations about the main variables pushing mothers to ask for help are not clear and not especially significant. The article need a very important revision to clarify 

The quality of the English is fully acceptable.

Author Response

According to the suggestions and comments of Reviewer 3 we made the following amendments.

Reviewer 4 Report

The manuscript “Female labor participation and mother’s daycare in Greece pre- and post-covid19 pandemic: A regional analysis” focuses on an important recent theme and is written in a very concise and clear way. For it to be considered for publication, some improvements should be made in the Introduction, Materials and Methods, Findings, Discussion, and Conclusion. 

The introduction is quite concise and clear. However, it should end by recalling the aim of the study and highlighting the hypotheses raised.

Section 3, Findings and discussion, I would prefer to see the results and discussion in separate sections.

In the Findings section, the results should be presented as clearly as possible. To make the description of the results clearer, I suggest that the test values be included in the body of the text especially when the effects are significant. I know that the values are all in the table, but it would look better if they were also in the text, to help the reader.  

As it stands, there is no real discussion of the results in light of the theory. The Discussion should also revolve around the hypotheses, making it clear whether these have been verified.

In the conclusions section, it would be good if the limitations of the study and suggestions for future studies were also highlighted.

Finally, references must be in accordance with the norms of the journal.

Author Response

According to the suggestions and comments of Reviewer 4 we made the following amendments,

Round 2

Reviewer 1 Report

The authors have taken action as required

Reviewer 2 Report

In the abstract, please remove ''don't'' and use ''do not''

The theory guiding the panel data approach has not been discussed. ANOVA technique is a method not a theory

Methodological gaps have not been discussed.

Contributions and policy recommendations should be discussed

I recommend the acceptance of the manuscript once the authors consider the above-mentioned comments

It is clear 
